# Identification of Novel microRNA Prognostic Markers Using Cascaded Wx, a Neural Network-Based Framework, in Lung Adenocarcinoma Patients

**DOI:** 10.3390/cancers12071890

**Published:** 2020-07-14

**Authors:** Jeong Seon Kim, Sang Hoon Chun, Sungsoo Park, Sieun Lee, Sae Eun Kim, Ji Hyung Hong, Keunsoo Kang, Yoon Ho Ko, Young-Ho Ahn

**Affiliations:** 1Department of Molecular Medicine, College of Medicine, Ewha Womans University, Seoul 07804, Korea; jeongseonkim821@gmail.com (J.S.K.); hellosieun@gmail.com (S.L.); kse096@naver.com (S.E.K.); 2Inflammation-Cancer Microenvironment Research Center, College of Medicine, Ewha Womans University, Seoul 07804, Korea; 3Division of Oncology, Department of Internal Medicine, College of Medicine, The Catholic University of Korea, Seoul 06591, Korea; rowett@catholic.ac.kr (S.H.C.); jh_hong@catholic.ac.kr (J.H.H.); 4Deargen, Inc., Daejeon 34051, Korea; sspark@deargen.me; 5Department of Microbiology, College of Natural Sciences, Dankook University, Cheonan 31116, Korea; kangk1204@gmail.com; 6Cancer Research Institute, College of Medicine, The Catholic University of Korea, Seoul 06591, Korea

**Keywords:** microRNA, lung adenocarcinoma, prognosis, Cascaded Wx, machine learning

## Abstract

The evolution of next-generation sequencing technology has resulted in a generation of large amounts of cancer genomic data. Therefore, increasingly complex techniques are required to appropriately analyze this data in order to determine its clinical relevance. In this study, we applied a neural network-based technique to analyze data from The Cancer Genome Atlas and extract useful microRNA (miRNA) features for predicting the prognosis of patients with lung adenocarcinomas (LUAD). Using the Cascaded Wx platform, we identified and ranked miRNAs that affected LUAD patient survival and selected the two top-ranked miRNAs (miR-374a and miR-374b) for measurement of their expression levels in patient tumor tissues and in lung cancer cells exhibiting an altered epithelial-to-mesenchymal transition (EMT) status. Analysis of miRNA expression from tumor samples revealed that high miR-374a/b expression was associated with poor patient survival rates. In lung cancer cells, the EMT signal induced miR-374a/b expression, which, in turn, promoted EMT and invasiveness. These findings demonstrated that this approach enabled effective identification and validation of prognostic miRNA markers in LUAD, suggesting its potential efficacy for clinical use.

## 1. Introduction

Lung cancer ranks first among all cancer types in incidence and mortality, accounting for 18.4% of cancer-related deaths worldwide [1]. Despite the continued development of novel treatment methods, including targeted and immuno-oncology therapies, which have significantly improved the survival rate of lung cancer patients, the 5-year survival rate of these patients remains <20% [2]. Therefore, identification and validation of prognostic markers useful for screening patients most likely to respond to a given therapy are urgently needed.

Prognostic markers include gene mutations, single-nucleotide polymorphisms of genes or regulatory elements, as well as levels of proteins, mRNAs, and noncoding RNAs. In particular, advances in global transcriptome analysis have promoted attempts to exploit RNA-expression levels as prognostic markers. For example, the analysis of reverse transcription-polymerase chain reaction (RT-PCR) data obtained from 147 patients with non-small cell lung cancer (NSCLC), the most common type of lung cancer, identified a six-gene signature (*STX1A*, *HIF1A*, *CCT3*, *HLA-DPB1*, *MAFK*, and *RNF5*) as a prognostic marker of poor patient outcomes [3]. In another study, microarray data from formalin-fixed paraffin-embedded (FFPE) samples from 55 NSCLC patients revealed a 59-gene prognostic signature [4]. Microarray profiling of microRNAs (miRNAs) in 104 lung adenocarcinoma (LUAD; a major subtype of NSCLC) patient samples revealed that high *hsa-mir-155* and low *hsa-let-7a-2* expression correlated with poor patient survival [5]. These attempts were groundbreaking but not particularly successful, possibly due to their small sample sizes, inconsistent platforms, inappropriate feature-processing steps, or lack of reliably robust methods for effectively analyzing high-dimensional data [6].

miRNAs, as the most extensively studied noncoding RNAs, are small single-stranded RNAs (19–25 nucleotides in length) and endogenous suppressors of target genes [7,8]. Their sequences are complementary to the 3′-untranslated regions (3′-UTRs) of target mRNAs and bind to these regions through Watson–Crick base pairing. Perfect matched binding of miRNAs to 3′-UTRs leads to mRNA degradation, whereas imperfect matched binding leads to translational repression. By suppressing target gene expression or protein translation, miRNAs regulate diverse physiological and pathological conditions, including cancer [8]. miRNAs can either promote or repress cancer development and progression according to their target genes. Moreover, numerous miRNAs can act as oncogenes by negatively regulating tumor suppressors. For example, miR-21 expression is upregulated in colon cancer and promotes cell growth and invasion by repressing the tumor suppressor *PTEN* [9]. miR-183 is overexpressed in colon cancer and represses *EGR1*, which encodes a transcription factor that acts as a tumor suppressor, to promote tumor cell migration [10]. Conversely, tumor-suppressor miRNAs can inhibit tumorigenesis, epithelial-to-mesenchymal transition (EMT), and metastasis by suppressing oncogenes. In lung cancer, let-7 controls cellular proliferation by negatively regulating the *KRAS* oncogene [11]. Additionally, miR-200 family members suppress EMT, migration, invasion, and metastasis of lung cancer cells by directly repressing *ZEB1*, a gene encoding an EMT-inducing transcription factor [12]. These represent examples of attempts to use miRNAs as biomarkers for cancer detection, diagnosis, prognosis, and drug efficacy [13].

We recently developed a novel prognosis-associated feature-selection framework called Cascaded Wx (CWx), an artificial neural network-based algorithm that ranks features (genes) according to cancer patient survival by training neural networks with high- and low-risk cohorts in a cascading fashion [6]. We used CWx to analyze information for LUAD patients (*n* = 507) among transcriptome data from The Cancer Genome Atlas (TCGA; 20,501 genes) and demonstrated the superiority of CWx to other models for identifying prognosis-related genes. In the present study, we applied the CWx platform to analyze LUAD TCGA miRNA-expression data to identify miRNA features associated with LUAD patient survival. Combined with NanoString miRNA assays in FFPE lung tumor samples, we validated the efficacy of several miRNAs selected by CWx for use as prognostic markers to predict survival in LUAD patients.

## 2. Results

### 2.1. CWx Ranks miRNA Features Associated with Survival of LUAD Patients

The CWx framework, which was originally developed to predict prognostic marker genes (mRNAs) [6], was adapted to analyze miRNA-expression (miRNA sequencing) data from 192 LUAD patients obtained from TCGA (Figure 1A). Among a total of 809 miRNAs, 197 (CWx miRNAs) were identified and ranked according to their prognostic potential CWx scores (Figure 1B and Appendix A). To gain insight into the biological functions of these CWx miRNAs, we performed an Ingenuity Pathway Analysis (IPA) (Figure 1C). Canonical pathways that were significantly enriched in the CWx miRNA list were “cancer drug resistance by drug efflux” (*p* = 9.77 × 10^−6^), “Th1 pathway” (*p* = 3.21 × 10^−3^), “Th1 and Th2 activation pathway” (*p* = 1.06 × 10^−2^), and “regulation of the EMT pathway” (*p* = 1.55 × 10^−2^), suggesting that many of the CWx miRNAs are relevant to cancer development and progression, especially to cancer immunity and EMT.

### 2.2. MiR-374a and MiR-374b Are Poor Prognostic Markers in LUAD

Among the CWx miRNAs, we selected the two top-ranked miRNAs (miR-374a and miR-374b) for further investigation. First, using NanoString, we measured the expression levels of these miRNAs in FFPE tumor samples obtained from 180 surgically resected LUAD patients (Table 1 and Figure 2A). The expression levels of miR-374a and miR-374b varied across samples and correlated positively with each other (Spearman’s ρ = 0.220; *p* = 0.003). To investigate the effects of miR-374a and miR-374b on the survival of LUAD patients, samples were divided into two groups according to miR-374a- or miR-374b-expression levels. Of these 180 patients, high miR-374a and miR-374b levels were detected in 33.3% and 59.4% of patients, respectively. High expression of miR-374a was associated with advanced pathological stage, and miR-374b was associated with smoking status (Table 1; *p* = 0.059 and *p* < 0.001, respectively). Additionally, Kaplan–Meier survival analysis demonstrated that high miR-374a expression tended to be associated with shorter recurrence-free survival (RFS) and overall survival (OS) (*p* = 0.059 and *p* = 0.082, respectively) of LUAD patients (Figure 2B and Appendix A). Moreover, patients with high miR-374b expression showed slightly shorter RFS than those with low miR-374b expression (*p* = 0.085), although there was no difference in OS between the two groups (*p* = 0.449; Figure 2B and Appendix A). We then assessed the risk factors affecting RFS and OS using the Cox proportional hazards regression method. We found that high miR-374a expression was related to poor RFS [hazard ratio (HR), 1.642; 95% confidence interval (CI): 0.995–2.709; *p* = 0.053) and OS (HR, 1.804; 95% CI: 1.084–3.003; *p* = 0.023) (Appendix A); however, the effect of miR-374b on patient survival was minimal and not statistically significant. Nevertheless, it should be noted that the combination of the two miRNAs (miR-374a and miR-374b) enhanced the prognostic value for LUAD (Figure 2C). Patients with high expression levels of both miR-374a and miR-374b (374a/b_high) showed shorter RFS than those with low expression levels of both miRNAs (374a/b_low; *p* = 0.049). Interestingly, miR-374a and miR-374b showed cancer-stage-specific effects on OS. In patients with stage I disease, high expression of miR-374a and miR-374b were associated with better OS (miR-374a, *p* = 0.012; and miR-374b, *p* = 0.002) (Appendix A), which is a similar finding to that reported previously [14]. By contrast, in patients with stage II LUAD, high expression of miR-374a and miR-374b showed a tendency toward poorer OS (miR-374a, *p* = 0.104; and miR-374b, *p* = 0.203) (Appendix A). These results suggested that the combination of miR-374a and miR-374b can be used as a prognostic marker in LUAD.

### 2.3. MiR-374a and MiR-374b Are Regulated by the ZEB1/miR-200 Feedback Loop

Epithelial cancer cells undergoing EMT acquire motility and invasiveness and finally metastasize to distant secondary sites [15]. Thus, EMT-related genes are closely associated with treatment response and survival of cancer patients [16,17]. miR-374a and miR-374b reportedly promotes EMT in breast [18], pancreatic [19], gastric [20], and lung cancer [21], whereas they also reportedly suppress EMT in bladder [22] and ovarian cancer [23]. To clarify the association between miR-374a/b and EMT, we measured levels of miR-374a and miR-374b in murine lung cancer cells in which EMT status had been manipulated by *ZEB1* and miR-200 [12]. *ZEB1* overexpression induces EMT in epithelial-like cells (393P), and *ZEB1* knockout or miR-200 overexpression suppresses EMT in mesenchymal-like cells (344SQ). In this system, both miR-374a and miR-374b levels were increased by *ZEB1* overexpression and decreased by *ZEB1* knockout or miR-200 overexpression (Figure 3A–C). Similar results were observed in human lung cancer cells, where *ZEB1* overexpression enhanced both miR-374a and miR-374b expression in HCC827 cells (Figure 3D), and use of the miR-200b mimic suppressed the expression of these miRNAs in A549 cells (Figure 3E). These results implied that miR-374a and miR-374b expression can be controlled by the *ZEB1*/miR-200 feedback loop, a potent EMT regulator.

### 2.4. MiR-374a and MiR-374b Promote EMT, Migration, and Invasion of Lung Cancer Cells

To investigate the effects of miR-374a and miR-374b on lung cancer cells, we transfected A549 human lung cancer cells with miR-374a and miR-374b mimics (Figure 4A). The miR-374a and miR-374b mimics slightly enhanced mRNA levels of *SNAI1* and *SNAI2* (mesenchymal markers) and inhibited those of *CDH1*, *INADL*, and *CRB3* (epithelial markers) (Figure 4B). Additionally, Western blot indicated that miR-374a and miR-374b enhanced ZEB1 and vimentin protein levels, indicating that miR-374a and miR-374b partially induced EMT in lung cancer cells (Figure 4C). In spheroid invasion assays, the miR-374a and miR-374b mimics promoted the invasion of A549 and H1792 cancer-cell spheroids into collagen gels (Figure 4D and Appendix A). Furthermore, miR-374a and miR-374b enhanced cancer cell migration in wound-healing assays (Figure 4E and Appendix A). These results suggested that miR-374a and miR-374b promote EMT, migration, and invasion of lung cancer cells.

### 2.5. MiR-374a and MiR-374b Induce Gene-Expression Signatures Related to EMT and Invasiveness

To analyze global changes in the transcriptome profile influenced by miR-374a and miR-374b, we performed RNA-sequencing analysis using miR-374a- and miR-374b-transfected A549 cells. Figure 5A shows the significantly upregulated (*n* = 169) or downregulated genes (*n* = 120; |fold change| ≥ 2, adjusted *p* ≤ 0.05, and average Transcripts Per Million ≥ 1) in both miR-374a- and in miR-374b-transfected cells relative to control cells. Rank–rank hypergeometric overlap analysis [24] revealed that two expression profiles (miR-374a vs. control and miR-374b vs. control) were significantly correlated (Figure 5B). We then uploaded the list of commonly upregulated or downregulated genes in both profiles (total: 289 genes) to the IPA system and explored the canonical pathways related to these genes. IPA analysis revealed canonical pathways, such as “ERK5 signaling (*p* = 1.48 × 10^−4^)”, “NGF signaling (*p* = 1.72 × 10^−3^)”, and “CD27 signaling (*p* = 2.82 × 10^−3^)” showing positive z-scores, suggesting that these pathways were activated in miR-374a/b-overexpressing cells. By contrast, “death receptor signaling (*p* = 3.45 × 10^−3^)” and “PTEN signaling (*p* = 4.33 × 10^−3^)” showed negative z-scores, suggesting that these pathways were inactivated in miR-374a/b-overexpressing cells (Figure 5C). To further analyze the signaling pathways involved in the molecular and cellular phenotypes affected by miR-374a and miR-374b, we performed Gene Set Enrichment Analysis (GSEA) and identified enriched or over-represented gene signatures in RNA-sequencing profiles (Figure 5D). As expected, “multi-cancer invasiveness” and “epithelial-mesenchymal transition” signatures were enriched in both datasets (miR-374a vs. control and miR-374b vs. control). We then selected several genes commonly included in the two signatures (*COL3A1*, *THY1*, *GREM1*, *SPOCK1*, *THBS2*, *TIPM3*, *COL5A1*, *SERR4*, and *COL11A1*) and measured their mRNA-expression levels in miR-374a/b-overexpressing A549 and H1792 cells (Figure 5E and Appendix A). The results indicated that the levels of most of the signature genes were elevated by miR-374a/b overexpression. This provided a molecular transcriptomic explanation for the observed miR-374a- and miR-374b-mediated EMT and invasiveness of lung cancer cells.

## 3. Discussion

Identification and validation of clinically applicable prognostic markers that accurately predict patient survival or drug response are crucial to achieving better treatment outcomes and improved survival rates in lung cancer. For this purpose, numerous studies have been conducted over the course of decades. Recently, one study demonstrated both circulating tumor cells exhibiting an active EMT status (vimentin+ and EGFR+) and tumoral expression of *AXL* mRNA as prognostic factors for OS and RFS of patients with early stage resectable NSCLC [25]. Using a different approach, we applied a neural network-based CWx framework to extract miRNA markers most highly associated with LUAD patient survival, profiled their expression levels in LUAD patient samples using NanoString technology, validated their effects on patient survival, and elucidated their functions in lung cancer cells. The results identified miR-374a and miR-374b, both EMT-related miRNAs, as potential prognostic markers associated with poor survival in LUAD patients.

Machine learning algorithms are useful for analyzing large volumes of data, such as genetic information produced by next-generation sequencing (NGS) technologies. Support vector machines [26], decision trees [27], and random forest [28] algorithms have been frequently adopted to extract prognostic features from high-throughput NGS profiling data [29,30,31,32]. Recently, we proposed that the CWx framework demonstrated enhanced feature-selection efficiency and increased accuracy in prognostic predictability as compared with previous algorithms [6]. Previous studies report that miRNA signatures show predictive, diagnostic, and prognostic value and are capable of enhancing the efficacy and feasibility of low-dose computed tomography screening in lung cancer patients [33,34]. These experimental results suggest that miRNAs can be used as effective lung cancer biomarkers. Therefore, we expanded the previously developed CWx framework for the analysis of miRNA profiling data and successfully extracted miRNA features associated with LUAD patient survival. Moreover, the top miRNAs identified by CWx were validated experimentally and clinically to demonstrate their prognostic potential.

miR-374 family members (miR-374a, -374b, and -374c) play indispensable regulatory roles in diverse physiological and pathological processes, including cancer development and metastasis [35]. In triple-negative breast cancer, miR-374a is upregulated relative to levels in non-tumor tissues and promotes cell proliferation, migration, and tumor progression in vivo by targeting arrestin-β1 (*ARRB1*) [36]. Additionally, miR-374a activates Wnt/β-catenin signaling by directly targeting *WIF1*, *PTEN*, and *WNT5A*, thereby promoting breast cancer metastasis [18]. In hepatocellular carcinoma cells, miR-374a promotes cell growth by targeting MIG-6 (*ERRFI1*), a negative regulator of EGFR signaling [23]. Moreover, miR-374b promotes cellular proliferation and inhibits apoptosis in gastrointestinal stromal tumors by targeting *PTEN* and activating PI3K/AKT signaling [37].

By contrast, miR-374a and miR-374b reportedly suppress the progression of some cancers. miR-374b suppresses the migration and invasion of bladder cancer cells by targeting *ZEB2*, an EMT-inducing transcription factor [22], and suppresses cell proliferation, migration, and EMT in ovarian cancer by targeting *FOXP1* [23]. Even in LUAD cells, miR-374a suppresses cell proliferation and invasion by targeting TGF-α (*TGFA*) [38], and in early stage NSCLC, high expression of miR-374a is associated with improved survival rates [14]. These conflicting roles of miR-374a and miR-374b were clarified in a study performed by Zhao et al. [39] in NSCLC cells, identifying dual stage-specific roles of miR-374a: suppression of cell growth, migration, invasion, and metastasis by targeting cyclin D1 (*CCND1*) in early-stage NSCLC while also targeting *PTEN* in advanced-stage NSCLC. Therefore, high miR-374a expression in early stage NSCLC is associated with improved patient survival rates, but in advanced NSCLC, it is associated with shorter survival, which is similar to the findings of the present study.

The CWx platform has predicted multiple other candidate miRNAs beyond miR-374a and miR-374b as associated with LUAD patient survival. Of these, let-7f is a member of the let-7 family, which includes well-known tumor-suppressor miRNAs that target oncogenes, such as *MYC*, *RAS*, and *CCND1* [40]. miR-101 inhibits lung cancer proliferation and metastasis by targeting *ZEB1* [41], and miR-200c is an EMT-inhibitory miRNA that also targets *ZEB1* [12]. Additionally, miR-21 is an oncogenic miRNA targeting *PTEN* and frequently upregulated in solid tumors [42]. These findings suggest that miRNA features derived through the CWx platform are related to cancer development and metastasis. Further studies are needed to validate the pathophysiological functions of these miRNA features in LUAD.

In summary, we conducted an integrated study that included machine learning, clinical sample profiling, and cellular experiments to predict and validate prognostic miRNA markers associated with LUAD patient survival. The results identified miR-374a and miR-374b as promoting cancer cell invasion through their elevated expression in LUAD patients and association with poor prognosis. We anticipate that the proposed CWx miRNAs will be useful as LUAD-specific prognostic markers following further experimental and clinical evaluation.

## 4. Materials and Methods

### 4.1. Data Acquisition

miRNA-expression data from 192 LUAD patients were obtained from TCGA via the firebrowse website (http://firebrowse.org/). These data contained normalized expression (reads per million) levels of 1,046 known miRNAs extracted from LUAD tissues. Of these miRNAs, 237 exhibiting no expression (or no changes between samples) were discarded. Therefore, expression values from a total of 809 miRNAs were used to extract core miRNAs associated with the prognosis of LUAD patients using the CWx algorithm.

### 4.2. CWx Analysis

The CWx algorithm was used to identify prognosis-associated miRNAs in 192 TCGA LUAD patients. First, patients were divided into high- (*n* = 98) and low-risk (*n* = 94) groups depending on whether they had survived for 3 years (Figure 1A). The number of miRNAs (features) was also reduced by ~50% at this step, after which the same process was conducted with different survival cut-offs (Figure 1A). Finally, 197 miRNAs were ranked according to CWx scores (the higher the CWx score, the more relevant to the prognosis).

### 4.3. Cell Culture

Cell lines 393P, 344SQ (murine lung cancer), A549, and H1792 (human lung cancer) were cultured in RPMI 1640 (Welgene, Gyeongsan, Korea) with 10% fetal bovine serum (FBS; HyClone, Logan, UT, USA) at 37 °C in the presence of 5% CO_2_. Murine lung cancer cells were established and transfected with *ZEB1* and miR-200 as described in our previous studies [12,43]. A549 cells were transduced with the pLMP-mCherry retroviral vector (a gift from Ken Scott, Baylor College of Medicine, Houston, TX, USA) to allow visualization via a red fluorescence signal. *hsa-miR-374a* mimic, *hsa-miR-374b* mimic, and negative controls were obtained from BIONEER (Daejeon, Korea) and transiently transfected into A549 or H1792 cells using TransIT-X2 transfection reagent (Mirus Bio, Madison, WI, USA) according to manufacturer instructions. For wound-healing assays, scratches were made with a 1000-µL pipette tip when cells became confluent in 6-well plates, and cells were cultured in complete media with 10% FBS in the presence of mitomycin C (1 µg/mL; Sigma-Aldrich, St. Louis, MO, USA) to block proliferation-related effects. After 24 h, the wound area was measured using Image J software (National Institutes of Health, Bethesda, MD, USA).

### 4.4. Quantitative RT-PCR (qRT-PCR)

We used WelPrep total RNA isolation reagent (Welgene) to isolate total RNA from cultured cells. To quantitate mRNA-expression levels, cDNA was first synthesized from total RNA by reverse transcription using the ELPIS RT Prime kit (Elpis-Biotech, Daejeon, Korea), and quantitative PCR assays was performed using a BioFACT A-Star real-time PCR kit including SFCgreen I (BioFACT, Daejeon, Korea) with the AriaMx real-time PCR system (Agilent Technologies, Santa Clara, CA, USA). mRNA levels were normalized to that of a housekeeping gene [ribosomal protein L32 (*RPL32*)]. qRT-PCR primers used in this study are listed in Appendix A. To quantitate cellular miRNA levels, we used an HB miR Multi Assay kit (Heimbiotek, Seongnam, Korea) and normalized miRNA levels to that of *RNU6B* snoRNA.

### 4.5. RNA Extraction from FFPE Tumors

LUAD patients (*n* = 180) who underwent surgical resection with a curative aim were retrospectively selected at Seoul St. Mary’s Hospital, Yeouido St. Mary’s Hospital, Bucheon St. Mary’s Hospital, or Uijeongbu St. Mary’s Hospital of Catholic Medical Center (Seoul, Korea). This study was approved by the institutional review board of Catholic Medical Center (No. UC17SESI0073). Total RNA was extracted from FFPE tumors from these patients using a miRNeasy FFPE kit (QIAGEN, Hilden, Germany) according to manufacturer instructions. RNA quantity and quality were assessed using a DS11 spectrophotometer (Denovix, Wilmington, DE, USA) and a fragment analyzer (Advanced Analytical Technologies, Ankeny, IA, USA).

### 4.6. MiRNA-Expression Profiling by NanoString

To measure miR-374a and miR-374b levels in FFPE samples, we performed an nCounter microRNA expression assay (NanoString Technologies, Seattle, WA, USA) using the human miRNA v3 assay kit by Philekorea (Seoul, Korea). Oligonucleotide-tagged miRNAs were hybridized with the human miRNA code set for 18 h at 65 °C, and the individual fluorescence intensity of target miRNAs was quantified using the nCounter digital analyzer, which was also used to obtain images of fluorescent reporters. miRNA data were normalized and analyzed using nSolver software (NanoString Technologies).

### 4.7. Western Blot

To isolate cellular proteins, cells were incubated with lysis buffer [50 mM Tris-HCl (pH 7.4), 150 mM EDTA, and 1% Triton X-100] containing protease inhibitors (Sigma-Aldrich). After electrophoresis (SDS-PAGE), proteins were transferred onto polyvinylidene difluoride (PVDF) membranes, and protein blots were incubated with primary antibodies and horseradish peroxidase-conjugated secondary antibodies (Cell Signaling Technology, Danvers, MA, USA). Protein bands were visualized with a PicoEPD (Enhanced Peroxidase Detection) Western reagent kit (Elpis-Biotech). We used antibodies against ZEB1 (#sc-25388; Santa Cruz Biotechnology, Dallas, TX, USA), vimentin (#sc-5565; Santa Cruz Biotechnology), and actin (#BS6007M; Bioworld Technology, St. Louis Park, MN, USA).

### 4.8. Spheroid Invasion Assay

Spheroid invasion assays were performed as described previously [44]. Briefly, to create spheroids, lung cancer cells (1 × 10^5^ cells/5 mL) in 20% METHOCEL (Sigma-Aldrich) and 1% Matrigel (BD Biosciences, Franklin Lakes, NJ, USA) were hung on the lid of 15 cm dishes (50 μL/drop) and incubated at 37 °C for 2 days. Spheroids were then harvested in a 15 mL tube, which was placed in the incubator for 30 min to allow the spheroids to settle. Spheroids were mixed gently with collagen solution (3 mg/mL collagen in 0.5× phosphate-buffered saline and 0.01 N NaOH) and then implanted in the center of each well of a 12-well plate. After the collagen gels polymerized, the wells were filled with cell culture media. After 24 to 48 h, invading cells were observed under a Leica DMi8 inverted microscope (Leica Microsystems, Wetzlar, Germany), and the invasion ratio was calculated by dividing the total invasion area by the central spheroid area measured using Image J software (National Institutes of Health).

### 4.9. RNA Sequencing

Total RNA was isolated from A549 cells transfected with miR-374a mimic, miR-347b mimic, or negative control in triplicate using an AccuPrep Universal RNA Extraction kit (BIONEER). NGS-based RNA sequencing for global mRNA transcriptome profiling was performed as described previously [44]. Briefly, after assessing the quantity and quality of RNA samples, a total RNA library was constructed using the Illumina TruSeq stranded mRNA sample prep kit (Illumina, San Diego, CA, USA). Indexed libraries were then submitted to Illumina NovaSeq (Illumina), and paired-end (2 × 100 bp) sequencing was performed by Macrogen (Seoul, Korea). Octopus-toolkit [45] was used to analyze RNA-sequencing data.

## 5. Conclusions

We analyzed LUAD TCGA miRNA data using CWx, an artificial neural network-based algorithm, and identified miRNA features associated with LUAD patient survival. Combined with NanoString miRNA assays using FFPE lung tumor samples, we identified and validated miR-374a and miR-374b as prognostic markers for predicting the survival of LUAD patients. Both miR-374a and miR-374b promoted EMT and the invasiveness of lung cancer cells through the induction of gene-expression signatures related to these phenotypes, suggesting their potential efficacy as LUAD prognostic markers.

## Figures and Tables

**Figure 1 cancers-12-01890-f001:**
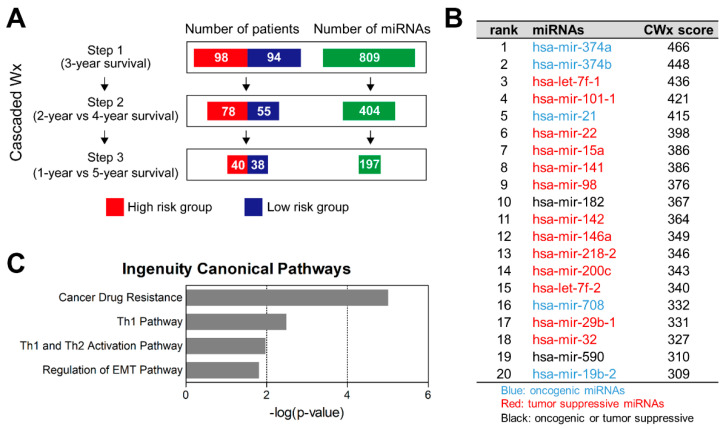
The CWx platform predicts miRNA features associated with lung adenocarcinomas (LUAD) patient survival. (**A**) Overview of the CWx miRNA-selection procedure. The number of samples (patients) was reduced through three cascading processes involving different criteria. Different cut-offs (3-year, 2- vs. 4-year, and 1- vs. 5-year) were used to categorize samples into high- or low-risk classes at the first, second, and the final steps, respectively. Additionally, the number of input miRNAs was reduced by ~50% at each step according to the prognostic potential (CWx score) of given miRNAs. (**B**) List of the CWx-miRNAs. The top 20 miRNAs predicted through the CWx platform are listed in the table along with their CWx scores. (**C**) IPA of the CWx miRNAs. CWx miRNAs (*n* = 197) were analyzed using the IPA tool to estimate enriched canonical pathways. Significantly enriched pathways (*p* < 0.05) are presented in the graph.

**Figure 2 cancers-12-01890-f002:**
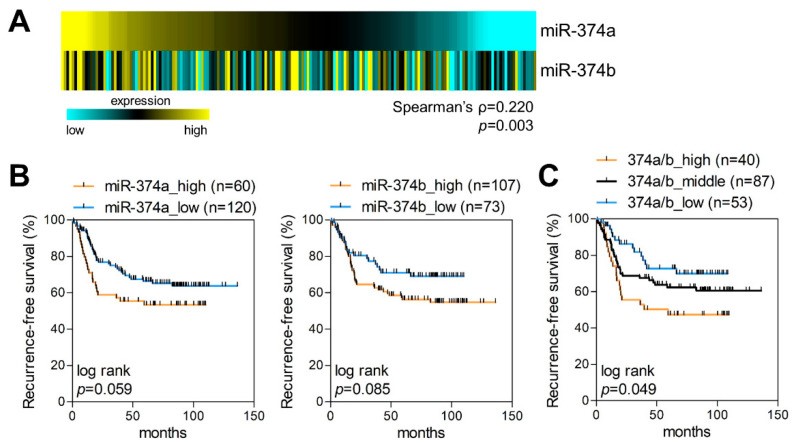
miR-374a and miR-374b are markers of poor prognosis in LUAD patients. (**A**) NanoString analysis of miR-374a and miR-374b in LUAD samples. miR-374a- and miR-374b-expression levels were measured in formalin-fixed paraffin-embedded (FFPE) tumors from LUAD patients (*n* = 180). (**B**) Recurrence-free survival (RFS) of LUAD patients based on miR-374a and miR-374b expression. LUAD patients were divided into two groups (high and low) based on expression levels of miR-374a (left) or miR-374b (right) obtained from NanoString. (**C**) RFS of LUAD patients based on miR-374a and miR-374b expression. LUAD patients were divided into three groups: 374a/b_high, high expression of both miR-374a and miR-374b; 374a/b_middle, low expression of one; and 374a/b_low, low expression of both miR-374a and miR-374b according to NanoString.

**Figure 3 cancers-12-01890-f003:**
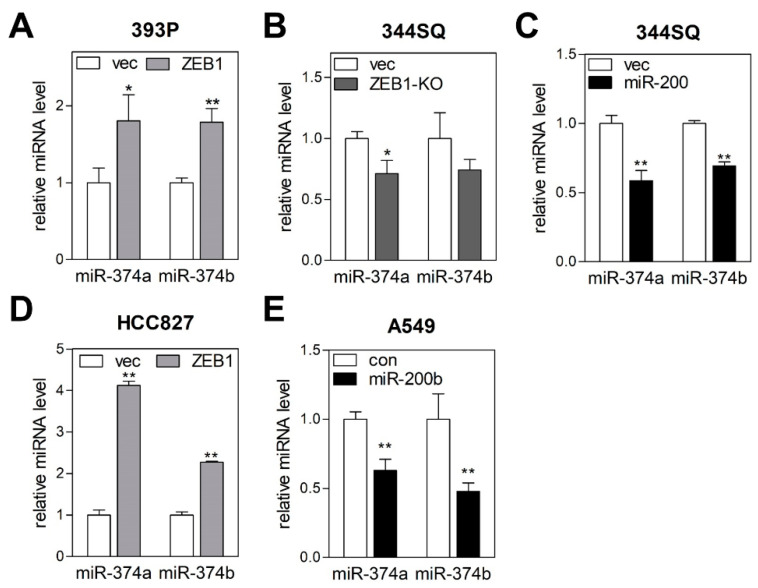
miR-374a and miR-374b are regulated by the *ZEB1*/miR-200 feedback loop. (**A**) qRT-PCR of miR-374a and miR-374b in *ZEB1*-overexpressing cells. 393P cells (epithelial-like murine lung cancer cells) were transfected with *ZEB1* or empty vector (vec), and miR-374a and miR-374b expression levels were measured by qRT-PCR. Expression levels were normalized to that of the small-nucleolar RNA *RNU6B*. Relative values to those of 393P_vec (set to 1.0) are presented. Data represent the mean + SD (*n* = 3). * *p* < 0.05, ** *p* < 0.01; two-tailed Student’s *t* test. (**B**) qRT-PCR of miR-374a and miR-374b in *ZEB1*-knockout (KO) cells. 344SQ cells (mesenchymal-like murine lung cancer cells) expressing Cas9 nuclease were transduced with *ZEB1* guide RNA (*ZEB1*-KO) or empty vector (vec), and miR-374a- and miR-374b-expression levels were measured by qRT-PCR. Data represent the mean + SD (*n* = 3). * *p* < 0.05; two-tailed Student’s *t* test. (**C**) qRT-PCR of miR-374a and miR-374b in miR-200-overexpressing cells. 344SQ cells were transduced with miR-200 or empty lentiviral vector (vec), and miR-374a- and miR-374b-expression levels were measured by qRT-PCR. Data represent the mean + SD (*n* = 3). ** *p* < 0.01; two-tailed Student’s *t* test. (**D**) qRT-PCR of miR-374a and miR-374b in *ZEB1*-overexpressing cells. HCC827 cells were transfected with *ZEB1* or empty vector (vec), and miR-374a- and miR-374b-expression levels were measured by qRT-PCR. Data represent the mean + SD (*n* = 3). ** *p* < 0.01; two-tailed Student’s *t* test. (**E**) qRT-PCR of miR-374a and miR-374b in miR-200b-transfected cells. A549 cells were transfected with miR-200b or negative control mimic (con), and miR-374a- and miR-374b-expression levels were measured by qRT-PCR. Data represent the mean + SD (*n* = 3). ** *p* < 0.01; two-tailed Student’s *t* test.

**Figure 4 cancers-12-01890-f004:**
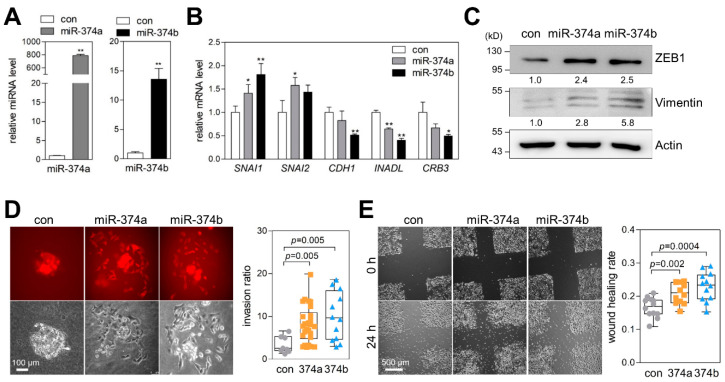
miR-374a and miR-374b promote epithelial-to-mesenchymal transition (EMT), migration, and invasion of lung cancer cells. (**A**) qRT-PCR of miR-374a and miR-374b in miR-374-mimic transfected cells. A549 cells were transiently transfected with miR-374a or miR-374b mimics, and miR-374a (left) and miR-374b (right) -expression levels were measured by qRT-PCR. Expression levels were normalized to that of the small-nucleolar RNA *RNU6B*. Relative values to those of A549 cells transfected with negative control (set to 1.0) are presented. Data represent the mean + SD (*n* = 3). ** *p* < 0.01; two-tailed Student’s *t* test. (**B**) qRT-PCR of EMT markers in miR-374-mimic transfected cells. A549 cells were transiently transfected with miR-374a or miR-374b mimic, and mRNA-expression levels of EMT markers were measured by qRT-PCR. Expression levels were normalized to that of *RPL32*. Relative values to those of A549 cells transfected with negative control (set to 1.0) are presented. Data represent the mean + SD (*n* = 3). * *p* < 0.05, ** *p* < 0.01; two-tailed Student’s *t* test. (**C**) Western blots of EMT markers in miR-374-mimic transfected cells. A549 cells were transiently transfected with miR-374a or miR-374b mimics, and protein levels of EMT markers were measured by Western blot. Actin was used as a loading control. Densitometric quantification is presented below the blots. (**D**) Spheroid invasion assay in miR-374-mimic transfected cells. Spheroids made from hanging-drop cultures of miR-374-mimic transfected A549 cells were seeded on collagen gels and then cultured for 24 h. A549 cells were labeled with mCherry by retroviral infection. Spheroid invasion ratios (ratio of whole cell area to the central spheroid area) were measured using ImageJ software. Box-and-whisker plots denote median and upper/lower quartiles + 1.5× interquartile range. Con, *n* = 8; 374a, *n* = 26; 374b, *n* = 12. *P*, two-tailed Student’s *t* test. Original magnification, 100×. (**E**) Wound-healing assay in miR-374-mimic transfected A549 cells. Cells were subjected to scratch wounds and then incubated for 24 h with mitomycin C (1 μg/mL) to block proliferation-related effect. Wound-healing rates [1−(wound area ratio of 24 h to that at 0 h)] were measured using ImageJ software (*n* = 12). *P*, two-tailed Student’s *t* test. Original magnification, 50×.

**Figure 5 cancers-12-01890-f005:**
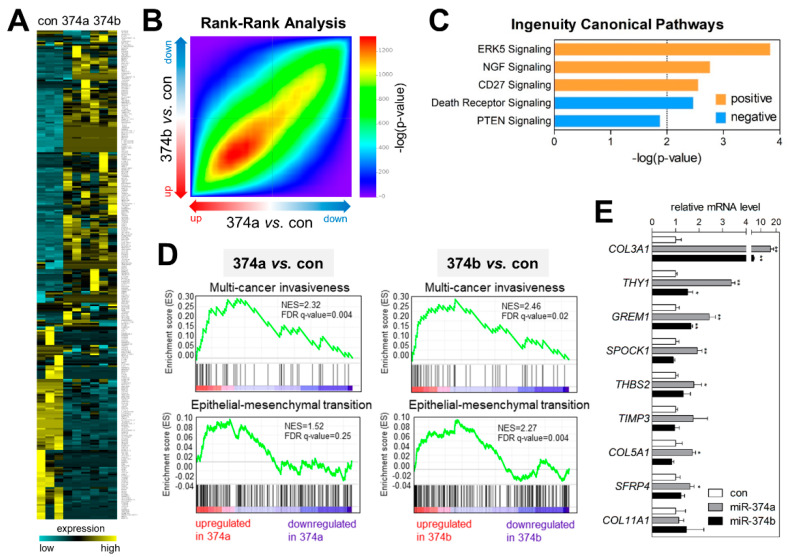
miR-374a and miR-374b induce gene signatures related to EMT and invasiveness. (**A**) Heatmap showing RNA-sequencing data obtained from miR-374a/b-overexpressing or control A549 cells. Commonly upregulated or downregulated genes in both miR-374a- and miR-374b-overexpressing cells as compared with controls are presented in the heatmap (|fold change| ≥ 2, adjusted *p* ≤ 0.05, and average Transcripts Per Million ≥ 1). Yellow: increased expression; blue: decreased expression. (**B**) Heatmap showing rank–rank hypergeometric overlap analysis of two profiling datasets (miR-374a vs. control and miR-374b vs. control). (**C**) IPA of genes differentially regulated by miR-374a/b. The top 500 commonly upregulated or downregulated genes in both profiles (a total of 1000 genes) were analyzed using the IPA tool to estimate enriched canonical pathways. The top five significantly enriched pathways are presented in the graph. (**D**) GSEA of differentially expressed genes in miR-374a- or miR-374b-overexpressing cells as compared with controls. The gene signatures “multi-cancer invasiveness” and “epithelial-mesenchymal transition” were enriched in both profiles. (**E**) qRT-PCR of EMT and invasiveness signature genes in miR-374-mimic transfected cells. A549 cells were transiently transfected with miR-374a or miR-374b mimics, and expression levels of the signature genes were measured by qRT-PCR. Expression levels were normalized to that of *RPL32*. Relative values to those in A549 cells transfected with negative control (set to 1.0) are presented. Data represent the mean + SD (*n* = 3). * *p* < 0.05, ** *p* < 0.01; two-tailed Student’s *t* test.

**Table 1 cancers-12-01890-t001:** Baseline characteristics of LUAD patients stratified based on miR-374a or miR-374b expression.

Variables	All Patients (*n* = 180)	*hsa-miR-374a-5p*	*hsa-miR-374b-5p*
Low (*n* = 120)	High (*n* = 60)	*p*	Low (*n* = 73)	High (*n* = 107)	*p*
Age, years	63.78 (28–85)			0.463			0.791
<65	94 (52.2%)	65 (69.1%)	29 (30.9%)		39 (41.5%)	55 (58.5%)	
≥65	86 (47.8%)	55 (64.0%)	31 (36.0%)		34 (39.5%)	52 (60.5%)	
Gender				0.916			0.312
Male	97 (53.9%)	65 (67.0%)	32 (33.0%)		36 (37.1%)	61 (62.9%)	
Female	83 (46.1%)	55 (66.3%)	28 (33.7%)		37 (44.6%)	46 (55.4%)	
Smoking				0.739			<0.001
Never smoker	101 (58.0%)	66 (65.3%)	35 (34.7%)		51 (50.5%)	50 (49.5%)	
Current + Ex-smoker	73 (21.3%)	29 (54.7%)	24 (45.3%)		22 (30.1%)	51 (69.9%)	
Differentiation grade				0.939			0.180
Well differentiated	49 (27.2%)	31 (63.3%)	18 (36.7%)		23 (46.9%)	26 (53.1%)	
Moderately	102 (56.7%)	70 (68.6%)	32 (31.4%)		41 (40.2%)	61 (59.8%)	
Poorly + Undifferentiated	29 (16.1%)	19 (65.5%)	10 (34.5%)		9 (31.0%)	20 (69.0%)	
Pathological stage				0.059			0.135
I	109 (61.2%)	77 (70.6%)	32 (29.4%)		47 (43.1%)	62 (56.9%)	
II	33 (18.6%)	22 (66.7%)	11 (33.3%)		14 (42.4%)	19 (57.6%)	
III + IV	36 (20.2%)	19 (52.8%)	17 (47.2%)		10 (27.8%)	26 (72.2%)	
Tumor size				0.134			0.510
<3 cm	106 (58.9%)	76 (71.7%)	30 (28.3%)		46 (43.4%)	60 (56.6%)	
3≤ T <7 cm	68 (37.8%)	40 (58.8%)	28 (41.2%)		24 (35.3%)	44 (64.7%)	
≥7 cm	6 (3.3%)	4 (66.7%)	2 (33.3%)		3 (50.0%)	3 (50.0%)	
Vascular invasion				0.907			0.852
No	152 (84.4%)	105 (69.1%)	47 (30.9%)		63 (41.4%)	89 (58.6%)	
Yes or unknown	28 (15.6%)	15 (53.6%)	13 (46.4%)		10 (35.7%)	18 (64.3%)	
Lymphatic invasion				0.758			0.889
No	114 (63.3%)	76 (66.7%)	38 (33.3%)		48 (42.1%)	66 (57.9%)	
Yes or unknown	66 (36.7%)	44 (66.7%)	22 (33.3%)		25 (37.9%)	41 (62.1%)	
Perineural invasion				0.969			0.949
No	169 (93.9%)	115 (68.0%)	54 (32.0%)		71 (42.0%)	98 (58.0%)	
Yes or unknown	11 (6.1%)	5 (45.5%)	6 (54.5%)		2 (18.2%)	9 (81.8%)	
EGFR mutation				0.104			0.737
No or unknown	116 (64.4%)	77 (66.4%)	39 (33.6%)		45 (18.9%)	71 (45.6%)	
Yes	63 (35.6%)	42 (66.7%)	21 (33.3%)		28 (33.3%)	35 (2.2%)	
Disease recurrence				0.128			0.053
No	113 (62.8%)	80 (38.8%)	33 (61.2%)		52 (46.0%)	61 (54.0%)	
Yes	67 (37.2%)	40 (44.4%)	27 (55.6%)		21 (31.3%)	46 (68.7%)	
Disease survival				0.155			0.407
Survival	115(63.9%)	81(70.4%)	34(29.6%)		44(38.3%)	71(61.7%)	
Death	65(36.1%)	39(60.0%)	26(40.0%)		29(44.6%)	36(55.4%)

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
