# Peer review of "Identification of Novel microRNA Prognostic Markers Using Cascaded Wx, a Neural Network-Based Framework, in Lung Adenocarcinoma Patients"

_cancers, 2020, doi:10.3390/cancers12071890_

Round 1

Reviewer 1 Report

Authors has withdrawn and re-submitted this paper. I would like to thank authors for addressing most of my previous comments. The addition of information about miRNA in the introduction section, adding more results for the evidence of miRNA prognostic markers using Cascaded Wx in LUAD has increased the quality of the paper. I would like to recommend authors to address few minor issue.

  1. Section 4.3 “Cell culture” Wound healing assay: Did author use 10% FBS for wound healing assay? 10% FBS can induce the migration of A549 cells. Please mentioned clearly. 

Also, author use mitomycin C to block the proliferation-related effect. In figure 4E, there are 3 groups (control, miR-374a, miR-374b), does this mean mitomycin was added to all these 3 group? Is there any group with just media control (without mitomycin C and miRNA) to confirm the real migration happening?

  1. Authors should include “fold change” densitometric calculation for western blot (figure 4C) to see significant difference.
  2. Figure 4 E legend, wound healing assay what was the magnification for microscopic image?
  3. There are still few grammatical errors.

Reviewer 2 Report

The authors have addressed most of my comments and the manuscript is much improved. The analysis data is clearer now and the description of the methods too. As far as I can see they have also addressed most of the point raised by the other reviewer. I consider that this manuscript fulfill the criteria to be published in Cancers.

Author Response

We really appreciate the reviewer’s positive comment.

This manuscript is a resubmission of an earlier submission. The following is a list of the peer review reports and author responses from that submission.

Round 1

Reviewer 1 Report

The manuscript reports on a study evaluating novel microRNA prognostic markers in lung adenocarcinoma patients identified through a neural network-based framework. This is a very nice study and certainly deserves further investigation.

Some comments:

  • The role of EMT has been recently investigated in early stage NSCLC using a different approach (de Miguel-Pérez D, et al. Cancers 2019). The results of this study should be included in the Discussion.
  • In early stage NSCLC, the use of two miRNAs classifiers has been reported as a potential minimally-invasive biomarker for lung cancer screening [Sozzi G, et al. J Clin Oncol 2014; Montani F, et al. J Natl Cancer Inst 2015]. The results of these studies paved the way to prospectively investigate the role of these miRNA classifiers coupled with LDCT screening. The inclusion of these studies in the Discussion would be useful for readers less familiar with the topic in order to understand the potential pleiotropic roles of miRNAs in lung cancer
  • I suggest to include a graphical abstract to briefly describe the study design and major findings

Reviewer 2 Report

General comments and concerns
The authors used their proprietary 'Cascaded Wx (CWx) platform' to explore the TCGA database for miRNAs involved in the prognosis of lung adenocarcinoma. The resulting two candidates were selected, miR-374a and miR-374b. They argue that miR-374a and miR-374b are prognostic predictors based on patient prognostic analysis and functional analysis using A549.

Although the method of screening is highly original and interesting, the method has already been reported by the authors themselves (Shin, B., et al. Front Genet 2019; 10: 662.) and the novelty of the method itself is limited.

Functional analysis of miR-374a and miR-374b was performed on only one cell line (A549 cells), and there is insufficient scientific evidence presented to claim that these miRs induce EMT. In addition, the novelty is limited because it has been reported that miR-374a produces EMT (Cai J., J Clin Invest. 2013;123:566.)

These two micro RNAs has not been investigated as clinical markers in patients with lung adenocarcinoma by evaluating recurrence free survival. This point is novel and interesting. However, the results are not statistically significant and therefore remain suggestive.

Specific points
1. In Figure 2B, the Y-axis starts at 50%. This graph is not scientifically problematic, but it is not often seen in clinical papers. The results seemed to be over-emphasized and confusing in my opinion.

2. In Figure 2B, the main statement of this paper is that miR-374a and miR-374b are prognostic predictors of lung adenocarcinoma. Figure 2B directly illustrates the phenomenon, but there is no statistically significant difference in the univariate analysis. It would be better to increase the number of cases or take care not to overstate the result.

3. In Fig. 2B, you stated two prognostic biomarkers in LUAD, miR-374a and miR-374b. Did you combined these two miRs to analyze reurrence free survival?

4. In Figure 2B, only univariate analysis is performed, but a multivariate analysis using COX proportional hazards model and other methods should be performed.

5. In Fig. 3, you use murine adenocarcinoma in your experiment. To prove the intent of this paper, it would be better to use two cell lines derived from homo sapiens.

6. In Figures 4 and 5, the functional analysis of miR-374a and miR-374b is performed. As mentioned above, these have only been performed on A549 cells and the scientific reproducibility is unknown. It is better to do a replicated experiment in one more cell line originated from human adenocarcinoma.

7. In Figure 4, some EMT markers were confirmed, but it is better to confirm protein expression as well as mRNA expression.

8. In Figure 4, the EMT phenotype is evaluated using only the invasion assay. It is a popular method, but it is also an ambiguous evaluation system. Then it is better to add another experiment to evaluate EMT phenotype and check the consistency. (Ex. wound healing assay, etc.).

Reviewer 3 Report

In this original article, authors have identified miRNA as a prognostic marker for lung adenocarcinoma. Lung cancer is still lacking effective prognostics markers leading to difficulty in its management at early stage. This urge for investigation of novel prognostic markers. This study is quite interesting and much need in the field of lung cancer. The results presented in this manuscript provides promising evidence as prognostic markers for LUAD. However, current manuscript needs major revision before considered appropriate for publication.

Major comments

  1. Introduction is too short. authors need to elaborate more about the microRNA, their role in cancer progression. E.g promotion of proliferation or metastasis/invasion of cancer cells.
  2. A cell free miRNA as a prognostic marker (e.g miRNA in serum/plasma, saliva, sputum) is more practical than measuring miRNA in FFPE sample. For FFPE, we need tissue sample and it is difficult and uncomfortable to both patient and oncologist to get lung tissue sample compared to withdraw blood and check. Moreover, serum-based or non-invasive methods (saliva, sputum) marker is easier and less time consuming than dealing with FFPE sample. Authors need to justify why FFPE would be advantage?
  3. It would be more interesting if author could categorize the miRNA listed in figure 1B into those tumor suppressor or tumor inducer (proliferation and migration) into a table if relevant. This will also relate the explanation mentioned in discussion line 251-270.
  4. Spheroid invasion assay needs to be elaborated more. How did author hung the A549 cells in lid of 15cm dish? How did author take image of spheroid: Cutting the spheroid section with microtome? Or directly the whole spheroid?
  5. Why is mCherry fluorescence color intensity of control, miR-374a and miR-374b different? Is it a problem with exposure? The migrated A549 cells in miR-374a group looks better in exposure compared to control and miR-374b. Please present different group with similar exposure.
  6. There are plenty of grammatical errors. Text are cryptic in many places. Authors need to check throughout the manuscript carefully.
  7. Few sentences (specially in methods section) is plagarised from following publication https://onlinelibrary.wiley.com/doi/abs/10.1002/ijc.32372. Authors need to re-phrase the sentence to decrease % similarity.
  8. Figure 4c legend. Magnification of image is missing

Minor comments

  1. Some words need to be italic font.
  2. Full form of some of the abbreviation is missing.